# Antitumor Effects of Resveratrol Opposing Mechanisms of *Helicobacter pylori* in Gastric Cancer

**DOI:** 10.3390/nu16132141

**Published:** 2024-07-04

**Authors:** Daniela Trautmann, Francesca Suazo, Keila Torres, Layla Simón

**Affiliations:** 1Nutrition and Dietetic School, Universidad Finis Terrae, Santiago 7501015, Chile; dtrautmannq@uft.edu (D.T.); fsuazom@uft.edu (F.S.); 2Department of Hematology and Oncology, School of Medicine, Pontificia Universidad Católica de Chile, Santiago 8331150, Chile

**Keywords:** polyphenol, natural product, antioxidant, anti-inflammatory, antitumor, gastric carcinoma

## Abstract

Gastric cancer is an aggressive and multifactorial disease. *Helicobacter pylori* (*H. pylori*) is identified as a significant etiological factor in gastric cancer. Although only a fraction of patients infected with *H. pylori* progresses to gastric cancer, bacterial infection is critical in the pathology and development of this malignancy. The pathogenic mechanisms of this bacterium involve the disruption of the gastric epithelial barrier and the induction of chronic inflammation, oxidative stress, angiogenesis and metastasis. Adherence molecules, virulence (CagA and VacA) and colonization (urease) factors are important in its pathogenicity. On the other hand, resveratrol is a natural polyphenol with anti-inflammatory and antioxidant properties. Resveratrol also inhibits cancer cell proliferation and angiogenesis, suggesting a role as a potential therapeutic agent against cancer. This review explores resveratrol as an alternative cancer treatment, particularly against *H. pylori*-induced gastric cancer, due to its ability to mitigate the pathogenic effects induced by bacterial infection. Resveratrol has shown efficacy in reducing the proliferation of gastric cancer cells in vitro and in vivo. Moreover, the synergistic effects of resveratrol with chemotherapy and radiotherapy underline its therapeutic potential. However, further research is needed to fully describe its efficacy and safety in treating gastric cancer.

## 1. Introduction

Gastric cancer is a prevalent malignancy and constitutes the fourth cause of cancer-related mortality globally, exhibiting a median survival period of only 12 months in its advanced stages [1]. The progression of gastric cancer occurs over an extended period, sometimes developing after 20–30 years of exposure to carcinogenic agents, and evolving through different stages [1,2]. Initially, non-cancerous cells undergo abnormal growth leading to dysplasia. Subsequently, normal cells are transformed into cancer cells, developing carcinoma in situ, in which cancer is confined to the superficial mucosal layer of the stomach. Finally, cancer cells spread through the different layers of the stomach and affect other nearby organs, resulting in invasive carcinoma [2].

Gastric cancer is an aggressive and multifactorial disease that includes environmental and genetic risk factors. Environmental factors, more than genetic alterations, increase the possibility of developing gastric cancer. For instance, dietary habits with a high intake of fruit and vegetables are protective, whereas processed meals, smoked meat and salt-preserved food are risk factors. In addition, smoking and alcohol intake are associated with a high risk of gastric cancer. For that reason, early diagnosis, proper treatment and prevention with diets are alternative therapeutic efforts to reduce patient suffering [1].

Infection with *Helicobacter pylori* (*H. pylori*) is a major environmental risk factor for gastric cancer and is linked to the development of gastric ulcers. This association is largely attributed to the inflammatory response induced by the infection with this bacterium [3]. Conventional therapy involves proton pump inhibitors and antibiotics. However, the indiscriminate use of antibiotics has led to microbial resistance. As a result, new studies on alternative therapies for eradicating the bacterium have been proposed. These novel therapies consider sterols and polyphenols as potential inhibitors of the activity of *H. pylori* [3].

Resveratrol is a natural polyphenol present in nuts, apples, red fruits, black olives, grapes and red wines. This natural product has antioxidant, anti-inflammatory and anti-microbial properties due to its ability to interact with reactive oxygen species (ROS) and cytoplasmic and nuclear proteins within cells. For that reason, resveratrol has been used as an alternative and complementary therapy for metabolic and cardiovascular diseases [4] and has also been effectively used in treating gastric cancer [5]. This review aims to provide a comprehensive summary of resveratrol as a potential therapeutic alternative in the prevention and treatment of gastric cancer induced by *H. pylori*.

## 2. *Helicobacter pylori*

*H. pylori* is a Gram-negative bacterium that colonizes the human stomach and is associated with various gastrointestinal diseases such as peptic ulcer disease and gastric cancer. *H. pylori* infection is one of the most common bacterial diseases worldwide, with an estimated 4.4 billion individuals infected. Almost 50% of the population over the age of 50 are infected with the bacterium, though prevalence varies across different geographical regions and ethnic groups [6,7,8,9].

The transmission of this infection occurs via oral–oral or fecal–oral routes. The pathogen exhibits a repertoire of mechanisms that enable it to colonize the stomach. Among these mechanisms are its motility, its adherence to gastric epithelial cells, and the modulation of the gastric environment [3]. Moreover, *H. pylori* promotes inflammation and oxidative stress, facilitating infection and carcinogenesis.

### 2.1. Mechanisms of Tumorigenesis Induced by H. pylori

Although only some individuals infected with *H. pylori* will develop gastric cancer, the presence of the bacterium significantly increases the risk [10]. The mechanisms through which *H. pylori* can trigger the development of gastric cancer are associated with infection (adhesion and colonization), the disruption of the gastric epithelium, DNA damage, the alteration of repair mechanisms, the activation of the inflammatory cascade, oxidative stress, angiogenesis and metastasis [11]. Figure 1 summarizes these mechanisms and factors induced by *H. pylori* during colonization, infection and tumorigenesis within the gastric mucosa. For instance, virulence factors, adherence molecules and cytokines promote inflammation and angiogenesis (Figure 1, red box) which are related to the promotion of gastric cancer (Figure 1, light pink box) and mucosa infection (Figure 1, blue box). Moreover, γ-glutamyl transferase (GGT) and urease are colonization factors that induce oxidative stress and angiogenesis (Figure 1, yellow box), thereby also promoting gastric cancer (Figure 1, light pink box) and infection (Figure 1, blue box).

The presence of the bacterium activates the release of proinflammatory cytokines, such as tumor necrosis factor-alpha (TNF-α), transforming growth factor-beta (TGF-β) and interleukins (ILs), leading to the infiltration of inflammatory cells into the gastric tissue. In this sense, *H. pylori* induces a chronic inflammation that perpetuates the damage to the gastric mucosa and promotes abnormal cell proliferation by triggering the proliferation of inactive endothelium to synthesize new blood vessels, thereby promoting angiogenesis (Figure 1, red box) [12].

Adhesion molecules and virulence factors of *H. pylori* play a crucial role in its ability to colonize and produce damage in the stomach. Proteins such as the blood group antigen binding adhesin (BabA), sialic acid-binding adhesin (SabA), the adherence associated lipoprotein A (AlpA) and outer membrane porin HopQ and HopZ and facilitate *H. pylori* adherence to the gastric epithelium. Factors like the outer inflammatory protein (OipA), cytotoxin-associated gene A (CagA) and vacuolating cytotoxin A (VacA) induce cellular damage, promoting chronic inflammation and the development of gastric cancer (Figure 1, red and light pink boxes). Additionally, the activity of GGT contributes to oxidative stress, induces the release of pro-inflammatory proteins, and favors *H. pylori* pathogenicity and persistence in the acidic gastric environment (Figure 1, yellow box) [3,13,14,15].

In addition, the helix shape and flagellar motility of the bacterium are essential for the survival and colonization of the stomach (Figure 1, blue box). Moreover, the enzyme urease facilitates the conversion of urea into ammonia and carbon dioxide, establishing a nearly neutral microenvironment that safeguards the bacterium against gastric acidity. Meanwhile, carbon dioxide acts as a protective barrier against bactericidal agents. These mechanisms underline the complex relationship between *H. pylori* and the stomach environment, highlighting the diverse strategies the bacterium employs to ensure its survival and involvement in pathological processes for the host, including gastric cancer and duodenal ulcer (Figure 1, yellow box) [16,17,18,19].

### 2.2. H. pylori-Induced Gastric Cancer

Gastric cancer is a malignant disease that affects the stomach and is characterized by its ability to invade and destroy surrounding tissues [2]. In 2020, the global incidence of gastric cancer was documented at approximately 1.09 million new cases, with a total of 769,000 fatalities reported. In this sense, gastric cancer was one of the top five cancer types in terms of mortality [20].

Genetic and external factors can influence the alterations in tumor cells in gastric cancer. Among the latter are obesity, exposure to carcinogens, the consumption of alcohol and tobacco, radiation exposure, and dietary habits, such as a high intake of salt, smoked, and processed meals, coupled with a low intake of fruits and vegetables. Infections with viruses or bacteria, such as the Epstein–Barr virus or *H. pylori*, alterations in the gastrointestinal microbiota, gastroesophageal reflux disease, gastric ulcers, or previous gastric surgery, also play a role in gastric cancer development [2,21].

Infection with *H. pylori* in the gastric mucosa initiates a chronic inflammatory response, leading to gastritis, which can progress through a multi-step gastric tumorigenesis cascade, known as the Correa cascade [2,22]. The first step of this progression is atrophic gastritis, characterized by sustained inflammation that causes the loss of normal gastric glands. Furthermore, atrophic gastritis can lead to intestinal metaplasia, a process in which gastric epithelial cells lining the stomach transform into cells resembling those of the intestine, an adaptive response to chronic inflammation regarded as a precancerous stage. Intestinal metaplasia can progress to dysplasia, where metaplastic cells begin to exhibit dysplastic changes, i.e., precancerous alterations in their shape and organization. Classified as low- or high-grade, high-grade dysplasia is a direct step toward cancer. Adenocarcinoma represents the final and malignant form of gastric cancer, where cells have undergone oxidative stress, mutations, and transformations allowing for the uncontrolled growth and eventual invasion of neighboring tissues, leading to metastasis [10,23].

## 3. Treatments for Gastric Cancer

In the early stages of gastric cancer, endoscopic resection is the recommended technique for managing lesions with a low risk of lymph node metastasis. Minimally invasive surgical techniques are viable alternatives for early lesions unsuitable for endoscopic intervention [24]. In contrast, advanced stages of gastric cancer require more aggressive treatments such as surgery, chemotherapy and radiotherapy. These therapeutic modalities aim to control the spread of cancer and enhance the quality of life for patients [25].

### 3.1. Conventional Treatments for Gastric Cancer

Surgery is commonly employed for the removal of gastric tumors. While it can be curative in some cases of gastric cancer, it presents challenges in other cases, where achieving complete tumor resection is difficult. Moreover, it carries inherent risks and potential complications, such as infections, bleeding and impaired gastric function (Table 1) [26,27].

Chemotherapy is applied both before surgery (neoadjuvant) and after surgery (adjuvant) to reduce the tumor size or eliminate remaining cancer cells, respectively. The combination of Fluorouracil, Leucovorin (calcium folinic acid), Oxaliplatin and Docetaxel is successful in some patients with gastric cancer [27]. Although it can be effective in destroying cancer cells, it can also induce significant side effects, including nausea, vomiting, fatigue, hair loss and the suppression of the immune system. Furthermore, drug resistance may limit its long-term efficacy [28]. In this sense, molecular therapeutic targets such as anti-angiogenic factors are used in gastric cancer refractory cases. For instance, Apatinib is a tyrosine kinase inhibitor that targets vascular endothelial growth factor receptor 2 and prolongs the survival rates of patients with advanced gastric cancer for whom conventional chemotherapy fails [27,29]. Moreover, PD-1 inhibitors and chemo-anti-angiogenic agent combination therapy improve the effects of immunotherapy in metastatic gastric cancer [30].

Radiotherapy employs high-energy radiation to destroy cancer cells and reduce tumor size. Radiotherapy can be useful for controlling tumor growth, treating the bleeding from the primary tumor, and alleviating symptoms in patients with metastatic gastric cancer. However, radiotherapy can also harm healthy tissues near the treated area, leading to side effects such as nausea, diarrhea, loss of appetite and fatigue. Additionally, some gastric tumors may exhibit intrinsic resistance to radiotherapy [31].

### 3.2. Polyphenols as a Potential Treatment for Gastric Cancer

As shown in Table 1, conventional treatments for gastric cancer have some disadvantages such as complications, side effects and resistance. For that reason, looking for alternative therapies for the prevention and treatment of gastric cancer is mandatory. In this scenario, natural products, specifically, polyphenols, which can control inflammation and oxidative stress, emerge as candidates.

Bioinformatic analysis has described genes that contribute to inflammation and oxidative stress in gastric cancer (e.g., IL-2, IL-6, IL-8, IFN-γ and LOX). Moreover, treatment with a polyphenolic cocktail and maggot larvae reduces the expression of these markers in a mouse model of gastric cancer [32]. In addition, a phenolic-rich extract of the Cactaceae *Nopalea cochenillifera* attenuates gastric lesions by inhibiting inflammation and oxidative stress. Specifically, this extract, enriched in the bioactive compound isorhamnetin, reduces gastric lesions, inflammation (TNF-α and IL-1β levels), and oxidative stress (COX-2 and MDA levels) in rats when administered at 100 mg/kg [33]. Furthermore, 10 mg/kg of carvacrol, a dietary polyphenol from Lamiaceae plants, reduces inflammation (IL-1β, IL-6, TNF-α and TGF-β) and oxidative stress (oxidative stress index (OSI) = total oxidant status (TOS)/total antioxidant status (TAS)) in a rat model of gastric cancer [34].

Polyphenols are also efficient in the prevention and treatment of *H. pylori*-induced gastric cancer. For instance, ellagitannins (castalagin and vescalagin) from *Castanea sativa* impair the *H. pylori*-induced inflammation (IL-8 release and NF-κB signaling) in human gastric cells [35]. Furthermore, the extracts obtained from *Cistus × incanus* L. and *Castanea sativa* are rich in polyphenols such as tannins, procyanidins and flavonoids, which can reduce IL-8 and NF-κB induced by *H. pylori* [36]. Moreover, the pretreatment of cells and mice with silibinin mitigates the *H. pylori*-induced dysplasia, hyperplasia, inflammation (NF-κB and STAT3 activation) and oxidative stress (COX-2 and iNOS expression) [37]. Similarly, walnut polyphenol extracts suppress the activation of STAT3 induced by *H. pylori* infection in RGM-1 gastric mucosal cells, thereby inhibiting the inflammatory and mutagenic action of the bacterium [38].

## 4. Resveratrol

Resveratrol is a polyphenol found in the berry family, including mulberries (5 mg of resveratrol per 100 g), blueberries (0.38 mg per 100 g), and cranberries (1.92 mg of resveratrol per 100 g), as well as in grapes (0.24 to 1.25 mg per 160 g) and nuts, such as peanuts (1.12 mg of resveratrol per 100 g) and pistachios (0.11 mg of resveratrol per 100 g) (Figure 2). It is renowned for its anti-inflammatory and antioxidant properties and has also been reported to possess antimicrobial properties by inhibiting the growth of *Staphylococcus aureus*, *Escherichia coli*, *Candida albicans* and *H. pylori* strains [6,39,40,41].

The appropriate dosage of resveratrol varies depending on the use and specific medical condition. Typical doses range from 100 to 500 mg of resveratrol daily, though some studies have used higher doses [42]. Resveratrol can be administered in numerous supplements and fortified foods [43,44]. However, the oral bioavailability of resveratrol is low due to its rapid metabolism and elimination. Only 12% of digested resveratrol reaches the bloodstream and body tissues [39]. For instance, when comparing dietary-achievable (5 mg) or pharmacological (1 g) doses of resveratrol, the concentrations required in blood and tissues to prevent cell growth are difficult to achieve in humans (Figure 2) [43]. For this reason, pharmaceutical formulations have been developed to improve the bioavailability of resveratrol. Techniques such as encapsulation in controlled-release systems and combination with other ingredients are employed to enhance both its absorption and bioavailability [43,45].

The co-encapsulation of resveratrol and Coenzyme Q10 into nanoparticles with rhamnolipid improves the chemical stability and release of resveratrol after in vitro gastrointestinal digestion [46]. Moreover, the encapsulation in nanoparticles with zein and pectin increases the solubility and the bioaccessibility of resveratrol in gastric and intestinal phases in vitro [47]. Nanoparticles with zein and Tween increase resveratrol bioaccessibility to 84% after the intestinal phase of in vitro digestion, compared with the low bioaccessibility of free resveratrol (13%) [48]. In addition, resveratrol loaded in chitosan–gellan nanofibers can deliver an equivalent to 840 μM of resveratrol with increased antioxidant capacity (Figure 2) [49].

The solid dispersion of resveratrol within superporous hydrogels based on blends of chitosan and polyvinyl alcohol sustains the drug release over 12 h in simulated gastric fluid [50]. Furthermore, the solid dispersion of resveratrol within films prepared of starch and chitosan has an efficient delivery of resveratrol (80% over 12 h), cytotoxic activity against human gastric cells, and anti-inflammatory effects against macrophage-like cells [51]. A solid dispersion of resveratrol supported by Magnesium di hydroxide has better solubility in gastric simulation fluid compared to pure resveratrol. In a clinical trial, human subjects consuming 180 mg of pure resveratrol, or the formulation supported by Magnesium di hydroxide, had a maximum plasma concentration of resveratrol of 2.2 and 6.3 μM, respectively, demonstrating a better bioavailability with the formulation (Figure 2) [45].

### 4.1. Properties of Resveratrol

Some properties of resveratrol are attributed to regulating various molecular mechanisms which are currently subjects of ongoing research regarding its therapeutic effects. Resveratrol has been suggested to modulate the activity of several enzymes and transcription factors, such as sirtuins, NF-κB and insulin-like growth factor 1 (IGF-1). These mechanisms involved oxidative stress, inflammation and tumor progression (Figure 3, all the mechanisms, light pink box) [52,53]. For instance, resveratrol has anti-inflammatory properties by reducing NF-κB, COX-2 and cytokines, in an *H. pylori*-opposite way (compare Figure 1 and Figure 3, red boxes). In addition, resveratrol can scavenge free radicals and reduce oxidative stress, while *H. pylori* has the opposite effect (compare Figure 1 and Figure 3, yellow boxes). In this sense, resveratrol inhibits *H. pylori* infection by reducing inflammation and oxidative stress (Figure 3, blue box). Moreover, resveratrol inhibits signaling pathways associated with cancer, such as mitogen-activated protein kinase (MAPK), AMP-activated protein kinase (AMPK) and protein kinase C (PKC). Also, resveratrol induces apoptosis and cell cycle arrest, but prevents angiogenesis, thereby exerting an antitumor effect (Figure 3, light pink box).

Resveratrol plays an anti-inflammatory role in inhibiting the activation of pro-inflammatory transcription factors such as NF-κB and activator protein 1 (AP-1) (Figure 3, red box). Moreover, resveratrol blocks the production of inflammatory mediators, such as prostaglandins and leukotrienes, by inhibiting the activity of key enzymes involved in their synthesis, including cyclooxygenase-2 (COX-2) and lipoxygenase (LOX). Resveratrol decreases the production of pro-inflammatory cytokines, such as TNF-α, interleukin-1β (IL-1β) and interleukin-6 (IL-6), while increasing the production of anti-inflammatory cytokines, such as interleukin-10 (IL-10). Furthermore, resveratrol modulates the function of immune system cells, such as macrophages and lymphocytes. For instance, resveratrol inhibits macrophage activation and reduces the production of ROS and pro-inflammatory cytokines by these cells. By reducing oxidative stress, resveratrol also attenuates the inflammatory response associated with oxidative damage [39].

Resveratrol acts as a potent antioxidant (Figure 3, yellow box), neutralizing free radicals and reducing oxidative stress in cells. Moreover, resveratrol has been shown to protect against oxidative damage and cellular aging, which may have implications in preventing diseases associated with oxidative stress, such as cardiovascular, neurodegenerative diseases and cancer [52]. Resveratrol modulates several cellular signaling pathways involved in oxidative stress, such as the nuclear factor erythroid 2- related factor 2 (Nrf2). This pathway promotes the expression of antioxidant enzymes, thereby protecting cells against oxidative damage [54,55].

In this context, resveratrol protects against *H. pylori*-induced gastritis by suppressing inflammation and oxidative stress, inhibiting markers like IL-8, iNOS, NF-κB and Nrf2 (Figure 3, light blue box) [56]. Additionally, the treatment with 100 µM of resveratrol for 2 h reduces the levels of ROS and IL-8 in MKN-45 gastric cells infected with a CagA-positive *H. pylori* strain [57]. Resveratrol derivatives have antibacterial effects against *H. pylori* viability, biofilm formation and infection in an in vivo model using *Galleria mellonella* [58]. Chitosan nanoparticles containing resveratrol have bactericidal effects against *H. pylori,* also preventing biofilm and in vitro (AGS and MKN-74 cell lines) and in vivo infection (*Galleria mellonella* model) [59].

### 4.2. Properties of Resveratrol in Gastric Cancer

Resveratrol has a tumor-suppressing effect on several cancers, including gastric cancer [5]. Likewise, this molecule can trigger apoptosis in tumor cells (Figure 3, light pink box). This involves the activation of intracellular signaling cascades that lead to DNA degradation and the destruction of cancerous cells. Also, resveratrol reduces the proliferation of tumor cells by blocking cell cycle progression. Resveratrol interferes with the activity of certain key proteins that regulate the cell cycle (i.e., cyclin D1 and CDK4), resulting in a halt in cell growth [6,60,61] and showing efficacy in inhibiting cancer metastasis [62].

Additionally, resveratrol inhibits the formation of new blood vessels in the process of angiogenesis, which is essential for the supply of nutrients and oxygen to tumors. By blocking angiogenesis, resveratrol deprives tumors of the necessary resources for their growth and spread [63,64]. This polyphenol interferes with several cellular signaling pathways involved in the proliferation and survival of tumor cells. Resveratrol modulates the activity of proteins such as NF-κB, MAPK and AMPK, which are involved in cellular survival and growth (Figure 3, light pink box) [39].

Specifically, resveratrol has shown effects on gastric cancer [65] independent of *H. pylori* infection. In KATO-III and SNU-1 cells, resveratrol inhibits the activity of PKC, thereby reducing the growth of gastric adenocarcinoma cells via tumor suppression, cell cycle arrest, reduced proliferation and increased apoptosis (Figure 3, light pink box) [66,67]. In this sense, the exposure of SNU-1 cells to 100 μM resveratrol for 24 h induces cell death and cell cycle arrest [68]. Furthermore, 25 and 50 μM resveratrol inhibit the viability of gastric cancer cells, AGS, BGC-823 and SGC-7901, in a sirtuin1-dependent manner through the induction of senescence and cell cycle arrest [60]. Similarly, resveratrol has shown effects in an animal model of gastric cancer. Indeed, 40 mg/kg/day of resveratrol reduces gastric cancer development by decreasing the fractions of Ki67-positive proliferative cells in the tumor specimens of nude mice [6,60].

The treatment of human gastric adenocarcinoma SGC-7901 cells with 50–200 μM resveratrol for 48 h induces apoptosis and DNA damage attributed to an increased generation of ROS and cytotoxicity [69]. Wu et al. (2018) demonstrated that treating SGC-7901 cells with 200 μM resveratrol reduces viability, induces S phase arrest, and increases the Bax/Bcl-2 ratio. This suggests that resveratrol inhibits the proliferation of gastric cancer cells, induces apoptosis, and suppresses NF-κB pathway [70].

Moreover, the treatment of other gastric cancer cell lines, AGS and MKN45, with resveratrol results in reduced heparanase activity alongside increased superoxide dismutase activity, an antioxidant enzyme. The reduction in heparanase activity correlates with decreased NF-kB transcriptional activity and attenuates the invasion potential in gastric cancer cells [71]. Gao et al. (2015) reported that resveratrol effectively inhibits the invasion and metastatic potential of gastric cancer cells in vitro by targeting the hedgehog signaling pathway and suppressing epithelial-to-mesenchymal transition (EMT). These results suggest the potential of resveratrol as a promising therapeutic agent in the treatment of gastric cancer [72].

Likewise, pre-treatment with resveratrol effectively attenuates the enhanced motility of gastric cancer cells and reverses the EMT induced by gastric cancer-derived mesenchymal stem cells (GC-MSCs) within the tumor microenvironment. This effect is mediated through the inhibition of the Wnt/β-catenin signaling pathway and results in the downregulation of the expression and secretion of several pro-inflammatory cytokines and angiogenic factors, including IL-6, IL-8, MCP-1 and VEGF, at both the RNA and protein levels [73]. Resveratrol decreases HIF-1α protein levels induced by hypoxia in SGC-7901 cells, thereby mitigating the stimulation of cell proliferation, migrative and invasive capabilities, as well as the induction of EMT changes through Hedgehog pathway activation [74].

Su et al. (2022) reported that resveratrol inhibited miR-155-5p expression, a microRNA whose upregulation promotes the initiation and progression of the cell cycle. This inhibition subsequently modulated the expression levels of claudin 1, cyclin D1, c-Myc, Bcl-2 and caspase-3, thereby arresting cell cycle progression in gastric cancer cell lines [75]. Yang et al. (2022) demonstrated that resveratrol downregulated the expression of *MALAT1*, impacting proliferation, migration and invasion. Additionally, the regulation of the MALAT1/miR-383-5p/DDIT4 signaling pathway by resveratrol induces apoptosis in cell line SGC7901 [76].

Similarly, resveratrol inhibits proliferation, migration and invasion, while inducing apoptosis by inhibiting β-catenin and Bcl-2 but increasing Bax expression in MKN7 cells [77]. Also, resveratrol prevents IL-6-induced invasion of SCG-7901 and HSC-39 cells by reducing the Raf-MAPK signaling pathway and the metalloproteinases (MMPs) MMP2 and MMP9 expressions. In addition, resveratrol treatment (20 μM resveratrol intratumorally injected every four days for 3 weeks) inhibits metastasis in an in vivo model of HSC-39 cells injected into the tail vein of nude mice [78].

After therapy with 25 mg/kg/2d resveratrol for 20 days, the tumors, formed by SGC-7901 cells injected into BALB/c nude mice, reduced their volume and weight. These effects were attributed to the induction of apoptosis, a reduction in EMT, migration and invasion through the inhibition of Akt/mTOR, MAPK/ERK and Wnt/β-catenin pathways, and the modulation of lncRNA H19 [79]. Moreover, resveratrol prevents gastric intestinal metaplasia through the inhibition of the Akt pathway, and the activation of p-FoxO4 signaling. In this sense, the expression of p-FoxO4 is reduced in intestinal metaplasia compared with normal tissues [80] and suggests that resveratrol may prevent metaplasia in humans.

Resveratrol loaded in Mesoporous Silica Nanoparticles (Res-loaded MSN) exhibits superior suppressive efficacy on HGC-27 and AGS cells, as well as enhanced anticancer effects on gastric cancer in vivo, compared to treatment with resveratrol alone. These results suggest that mesoporous silica nanoparticles might be a promising drug delivery system for gastric cancer therapy [81].

Altogether, these findings suggest that resveratrol can control *H. pylori*-induced effects by suppressing infection, inflammation and oxidative stress. Additionally, resveratrol reduces the growth of cancer cells through several mechanisms, notably its role in signaling pathways, cell cycle, apoptosis and senescence. Given this, resveratrol represents a significant barrier against the initiation, development and progression of *H. pylori*-induced gastric cancer.

### 4.3. Combination Therapy Strategies with Resveratrol

Resveratrol has demonstrated synergy with various chemotherapeutic agents, such as 5-fluorouracil (5-FU) and doxorubicin, thereby enhancing their antitumor effects across several types of cancers [63,82]. Hu et al. (2019) reported the promising therapeutic potential of resveratrol, particularly when combined with ginkgetin, a biflavone, and 5-FU, resulting in the suppressed expressions of COX-2 and inflammatory cytokines and relieving the 5-fluorouracil-induced inflammatory response (Table 2). Likewise, ginkgetin and resveratrol inhibit VEGF-induced endothelial cell proliferation, migration, invasion and angiogenesis [63].

Resveratrol combined with doxorubicin regulates the SIRT1/β-catenin signaling pathway to reverse EMT, consequently inhibiting cell migration and promoting cell apoptosis, thereby enhancing their antitumor effects [82,83]. In an in vivo study involving nude mice with xenografts of doxorubicin-resistant subclones from a gastric cancer cell line (SGC7901), treatment with 3 mg/kg of doxorubicin and 50 mg/kg of resveratrol achieved a tumor volume reduction of 86.97%. Immunohistochemistry confirmed that the combination treatment significantly enhanced PTEN and caspase-3 expression while inhibiting vimentin and Ki67 expression, compared to individual treatments. Resveratrol can delay tumor growth by inhibiting EMT and promoting apoptosis, thereby reversing doxorubicin resistance in gastric cancer [84].

Resveratrol enhances the cytotoxic profile of Herceptin (trastuzumab) and exhibits synergistic interactions with it, resulting in a marked decrease in the gene expression of Bcl-xL, and thereby, the induction of apoptosis [85]. Kong et al. (2017) reported that the combination of resveratrol (5 μg/mL or 10 μg/mL) and paclitaxel (5 μg/mL) demonstrated synergistic growth inhibitory effects and increased apoptosis compared to separate treatments. This combinatory treatment significantly downregulates the mRNA and protein expressions of various key molecules involved in cancer progression, including COX-2, VEGF, MMP1, MMP2, MMP9, NF-κB, anti-apoptotic proteins (Bcl-2, Bcl-xL), procollagen I, collagen I, collagen III and CTGF, as well as pro-inflammatory cytokines (TNF-α, IL-1β) and iNOS, while upregulating the expression of anti-cancer genes (TIMPs—TIMP-1, TIMP-2, TIMP-3, IκB-α, p53, p21, caspases—caspase-3, caspase-8, caspase-9, Bax). These findings suggest that the combined treatment could serve as an effective enhancer of the anticancer properties of paclitaxel [86].

Türkmen et al. (2023) demonstrated that resveratrol effectively reduces both the expression of caspase-3 protein and the histopathological changes resulting from treatment with pembrolizumab (PEMB), a monoclonal antibody targeting the PD-1 surface protein and used as an anticancer agent. Moreover, resveratrol administration effectively mitigates the biochemical, immunological and histological alterations induced by PEMB [87]. A clinical trial revealed that the combination of resveratrol and copper reduces the toxicity of docetaxel chemotherapy in advanced gastric cancer. Specifically, resveratrol reduces the incidence of non-hematological toxicities such as hand–foot syndrome, diarrhea, and vomiting (13%) [88].

The cisplatin–resveratrol combination exhibits significant efficacy in suppressing metastasis. This treatment hinders proliferation and decreases cell invasion and telomerase activity while inducing ß-galactosidase activity. Additionally, the combination treatment leads to cell cycle arrest at the G0/G1 stage and promotes apoptosis and cell senescence by targeting the P38/P53 and P16/P21 pathways [89]. Additionally, resveratrol and cisplatin synergistically inhibit the proliferation of AGS gastric cancer cells by inducing endoplasmic reticulum stress-mediated apoptosis and G2/M phase cell cycle arrest [90].

Resveratrol safeguards against radiation-induced intestinal damage caused by radiotherapy by reducing oxidative stress and apoptosis through the SIRT1/FOXO3a and PI3K/Akt pathways in normal cells [91]. Moreover, resveratrol enhances the efficacy of radiotherapy by increasing the sensitivity of tumor cells to radiation and promoting apoptosis [64]. The radio-sensitizing effect of resveratrol is achieved by impairing DNA repair capacity in the early stages, inhibiting proliferation, inducing autophagy, and promoting apoptosis both in vitro and in vivo [92].

**Table 2 nutrients-16-02141-t002:** Synergistic effects of resveratrol with chemotherapeutic agents and radiotherapy.

Therapy Agent	Effects of Combination with Resveratrol	References
5-Fluorouracil (5-FU)	Mitigating the inflammatory response induced by 5-FU	[63]
Doxorubicin	Inhibiting cell migration and EMT, promoting cell apoptosis	[82,83]
Trastuzumab (Herceptin)	Induction of apoptosis	[85]
Paclitaxel	Growth inhibitory effects and increased apoptosis	[86]
Pembrolizumab (PEMB)	Mitigating the biochemical, immunological, and histological alterations induced by PEMB	[87]
Docetaxel	A combination of resveratrol and copper reduces the toxicity of docetaxel chemotherapy	[88]
Cisplatin	Suppressing metastasis, promoting apoptosis and cell senescence, inhibiting proliferation, and causing cell cycle arrest	[89]
Radiotherapy	Safeguarding against radiation-induced intestinal damage, reducing oxidative stress and apoptosis in normal cells, increasing the sensitivity of tumor cells to radiation, promoting apoptosis, and inducing autophagy	[91]

On the other hand, resistance mechanisms to resveratrol treatment involve several factors, including the activation of alternative signaling pathways or survival mechanisms by cancer cells, which can counteract the effects of resveratrol [93]. Additionally, the efficacy of resveratrol is influenced by the metabolism of the compound; variations in the expression of drug-metabolizing enzymes may significantly affect therapeutic outcomes [94]. Furthermore, the modulation of transport proteins within cell membranes plays a crucial role in regulating the absorption and excretion of resveratrol [95,96].

## 5. Conclusions

Researching resveratrol as a promising therapeutic option for *Helicobacter pylori*-induced gastric cancer provides new insights and avenues for treatment. With its antioxidant, anti-inflammatory and antitumor properties, resveratrol could offer therapeutic benefits without the side effects and toxicity associated with conventional therapies. However, further research and clinical studies are essential to thoroughly evaluate its efficacy and safety in the context of gastric cancer and to determine the optimal dosages, formulations, administration routes and drug interactions to fully harness the therapeutic potential of resveratrol in gastric cancer. Finally, future perspectives for resveratrol as a treatment for gastric cancer include the development of more comprehensive clinical studies to validate its therapeutic potential. With further evidence, resveratrol could become a more widely accepted component in clinical practice due to its antitumor properties.

## Figures and Tables

**Figure 1 nutrients-16-02141-f001:**
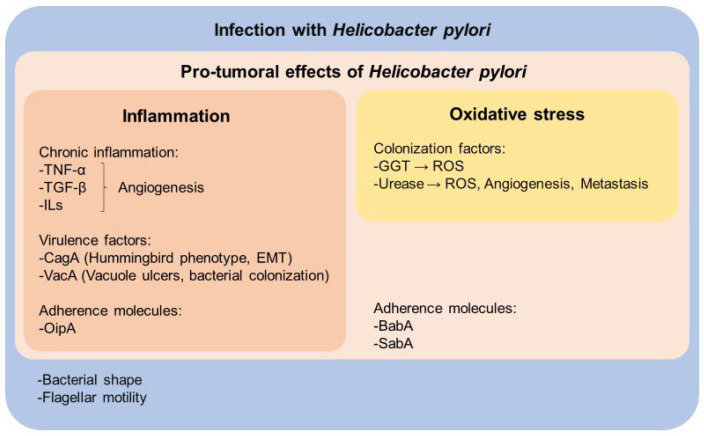
Mechanisms associated with *H. pylori* gastric infection (all, blue box) and the development of gastric cancer (light pink box). *H. pylori* presents adherence molecules and virulence factors such as CagA and VacA that alter the gastric epithelium. CagA reduces cell adhesion and promotes epithelial–mesenchymal transition (EMT). VacA induces bacterial incorporation into the host cell membrane and enhances bacterial colonization into the stomach. Additionally, gastric carcinogenesis is associated with inflammation, which induces angiogenesis (red box). Colonization factors such as GGT and urease promote the generation of reactive oxygen species (ROS), angiogenesis, and metastasis (yellow box).

**Figure 2 nutrients-16-02141-f002:**
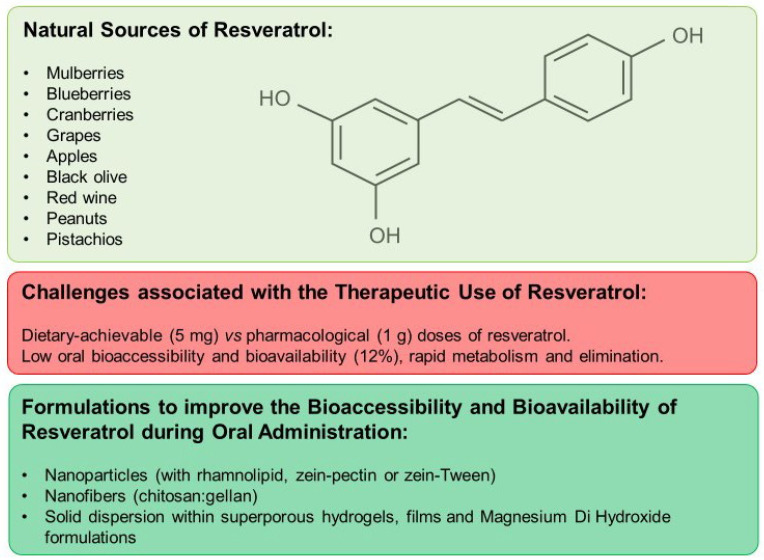
Resveratrol: natural sources, challenges associated with its therapeutic use, and formulations to improve its bioaccessibility and bioavailability during oral administration.

**Figure 3 nutrients-16-02141-f003:**
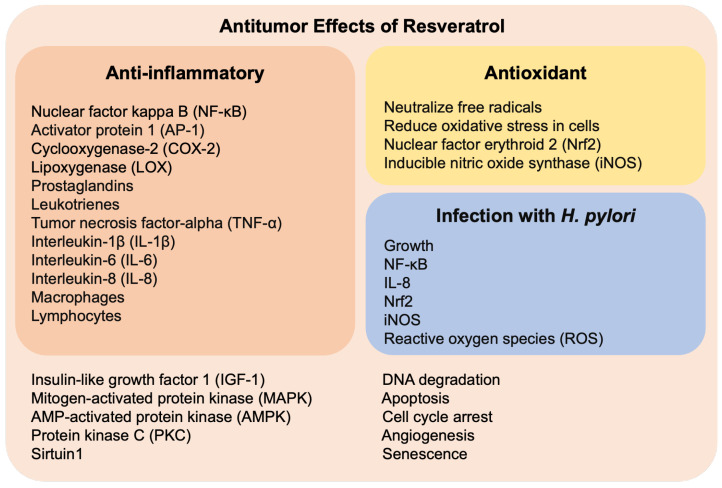
Mechanisms of resveratrol in preventing inflammation (red box), oxidative stress (yellow box), *H. pylori* infection (light blue box) and, thereby, cancer (all, light pink box).

**Table 1 nutrients-16-02141-t001:** Conventional Treatments for Gastric Cancer.

Treatment	Advantages	Disadvantages	References
Surgery	Curative in early stages of GC.	Complications: infections, bleeding, thrombus, impaired gastric function.	[26,27]
Chemotherapy: Fluorouracil, Leucovorin, Oxaliplatin and Docetaxel	Reduces tumor size before surgery. Eliminate remaining cells after surgery.	Side effects: nausea, vomiting, fatigue, hair loss, and suppression of the immune system. Drug resistance.	[27,28]
Molecular therapeutic agent: Apatinib, PD-1 inhibitors	Useful in refractory cases and metastatic gastric cancer.	Side effects: proteinuria and hypertension.	[27,29,30]
Radiotherapy	Metastatic GC. Useful as a palliative effect and for control of the bleeding from the primary tumor.	Side effects: nausea, diarrhea, loss of appetite, and fatigue. Resistance.	[31]

GC: Gastric cancer.

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
