# Peer review of "Antitumor Effects of Resveratrol Opposing Mechanisms of Helicobacter pylori in Gastric Cancer"

_nutrients, 2024, doi:10.3390/nu16132141_

Round 1

Reviewer 1 Report

Comments and Suggestions for Authors

In this comprehensive review, the author provides a thorough summary of the current knowledge surrounding Helicobacter pylori and resveratrol treatment in gastric cancer. The manuscript summarizes the molecular mechanisms underlying gastric cancer cells during Helicobacter pylori infection and discusses the potential clinical applications of resveratrol-mediated anti-tumor effects. This article highlights the importance of combining resveratrol with chemotherapeutic agents such as 5-FU and doxorubicin in the treatment of gastric cancer. To enhance the value of this review and contribute novel insights, the author should incorporate the following comments:

1.     The reviewer did not find a clear correlation between Helicobacter pylori and resveratrol treatment in gastric cancer. Although the manuscript described the molecular effects induced separately by Helicobacter pylori and resveratrol, the author should provide more correlated evidence to support this notion.

2.     Does Helicobacter pylori or resveratrol have any stage-specific effects during the progression of gastric cancer?

3.     The cumulative incidence of gastric cancer at 20 years after detection of Helicobacter pylori infection was 0.65%, suggesting this bacterium is only one possible cause of gastric cancer. Why should resveratrol treatment focus on Helicobacter pylori-infected gastric tumors? Please emphasize this information in the abstract.

4.     Resveratrol exerts tumor-suppression activity in gastric cancer cell lines including KATO-III, SNU-1, SGC-7901, AGS, and BGC-823. Are these cell lines infected with Helicobacter pylori? Highlight this information prominently in the text.

5.     More detailed information should be included in Figure 1. For example, Helicobacter pylori infection is shown to reduce cell adhesion and promote epithelial-mesenchymal transition and pro-inflammatory responses through CagA in gastric cells. Additionally, the VacA toxin causes incorporation into the host cell membrane, enhancing the bacteria's ability to colonize the stomach.

6.     Specific treatments and agents should be listed in Table 1. For instance, chemotherapy includes fluorouracil, leucovorin, and oxaliplatin? Molecular therapeutic agents could include PD-1 inhibitors.

7.     Consider using a table to highlight key information in the combination therapy strategies with resveratrol for improved readability and comprehension.

Comments on the Quality of English Language

Minor editing of English language required

Author Response

We hope this email finds you well. We sincerely thank you and the reviewers for reviewing our article entitled now “Antitumor Effects of Resveratrol Opposing Mechanisms of Helicobacter pylori in Gastric Cancer”. We greatly appreciate your insightful comments and suggestions, which have undoubtedly enhanced our work.

Below, we have addressed each of the observations provided (in blue) and its corresponding response (in black):

Reviewer #1:

  1. The reviewer did not find a clear correlation between Helicobacter pylori and resveratrol treatment in gastric cancer. Although the manuscript described the molecular effects induced separately by Helicobacter pylori and resveratrol, the author should provide more correlated evidence to support this notion.

Thank you for your comment. In this manuscript, we aim to highlight how resveratrol and H. pylori infection have opposing properties in the development of gastric cancer. While resveratrol possesses anti-inflammatory, antioxidant and antitumor properties, H. pylori infection promotes inflammation, oxidative stress and carcinogenesis. In Sections 2 and 4, we discuss the individual effects of H. pylori and resveratrol on gastric cancer, laying the groundwork for exploring their combined impact. In Section 4.1 “Properties of Resveratrol”, we mention that H. pylori infection triggers a robust inflammatory response and oxidative damage, both of which are key drivers in the pathogenesis of gastric cancer. By mitigating these effects, resveratrol interferes with the carcinogenic processes initiated by H. pylori. Particularly, we mention that resveratrol protects against H. pylori-induced gastritis by suppressing inflammation and oxidative stress, inhibiting markers like IL-8, iNOS, NF-κB and Nrf2. Moreover, we review the literature describing the effects of resveratrol over gastric cancer cells infected with H. pylori in in vitro and in vivo models [56-59]. In Section 4.2 “Properties of Resveratrol in Gastric Cancer,” we show that resveratrol's impact on gastric cancer cells includes reducing proliferation, inducing apoptosis, and suppressing invasive potential by modulating signaling pathways such as NF-κB, MAPK and Wnt/β-catenin.

  1. Does Helicobacter pylori or resveratrol have any stage-specific effects during the progression of gastric cancer? H. pylori initiates gastric carcinogenesis through mechanisms such as chronic inflammation, oxidative stress and disruption of the gastric epithelial barrier, leading to chronic gastritis, gastric atrophy and intestinal metaplasia, which are precancerous conditions. Virulence factors further promote angiogenesis and metastasis in advanced stages of cancer. Conversely, resveratrol exerts protective effects by reducing inflammation and oxidative stress, inhibiting cancer cell proliferation and preventing angiogenesis.

  1. The cumulative incidence of gastric cancer at 20 years after detection of Helicobacter pylori infection was 0.65%, suggesting this bacterium is only one possible cause of gastric cancer. Why should resveratrol treatment focus on Helicobacter pylori­infected gastric tumors? Please emphasize this information in the abstract.

We added in the abstract: “Although only a fraction of patients infected with H. pylori progresses to gastric cancer, bacterial infection is critical in the pathology and development of this malignancy.” Referring to resveratrol, we extended the sentence: “This review explores resveratrol as an alternative cancer treatment, particularly against H. pylori-induced gastric cancer, due to its ability to mitigate the pathogenic effects induced by the bacterial infection.”

  1. Resveratrol exerts tumor-suppression activity in gastric cancer cell lines including KATO-Ill, SNU-1, SGC-7901, AGS, and BGC-823. Are these cell lines infected with Helicobacter pylori? Highlight this information prominently in the text.

It is important to note that these cell lines are not typically infected with Helicobacter pylori in standard in vitro studies. We added in the text “Specifically, resveratrol has shown effects on gastric cancer independent of H. pylori infection”.

  1. More detailed information should be included in Figure 1. For example, Helicobacter pylori infection is shown to reduce cell adhesion and promote epithelial-mesenchymal transition and pro-inflammatory responses through CagA in gastric cells. Additionally, the VacA toxin causes incorporation into the host cell membrane, enhancing the bacteria's ability to colonize the stomach.

We added this information to Figure 1 and its legend.

  1. Specific treatments and agents should be listed in Table 1. For instance, chemotherapy includes fluorouracil, leucovorin, and oxaliplatin? Molecular therapeutic agents could include PD-1 inhibitors.

Thank you for your suggestion. We added this information in Table 1.

  1. Consider using a table to highlight key information in the combination therapy strategies with resveratrol for improved readability and comprehension.

To enhance readability and comprehension, we have designed a table (Table 2) to highlight key information on the combination therapy strategies with resveratrol.

Reviewer 2 Report

Comments and Suggestions for Authors

Dear authors

This review is very complete, the english language is also very good.

Resveratrol seems to be an interesting tool but gastric cancer treatment, but most studies are in rats.

There are some studies from 2017 in humans, but its acceptance and use in clinical practice is still waiting? Can you explain why? Can you give the future perspective of resveratrol use?

I would suggest to change the title, focusing more in resveratrol

Author Response

Reviewer #2:

  1. There are some studies from 2017 in humans, but its acceptance and use in clinical practice is still waiting? Can you explain why?

We appreciate the reviewer's insightful comment. The delay in the acceptance and use of resveratrol in clinical practice is primarily due to the limited number of large-scale, randomized controlled trials that can conclusively establish its efficacy and safety in humans.

  1. Can you give the future perspective of resveratrol use?

We added the following paragraph in the conclusion: “Finally, future perspectives for resveratrol as a treatment for gastric cancer include the development of more comprehensive clinical studies to validate its therapeutic potential. With further evidence, resveratrol could become a more widely accepted component in clinical practice due to its antitumor properties.”

  1. I would suggest to change the title, focusing more in resveratrol

Thank you for your suggestion. We considered changing the title to “Antitumor Effects of Resveratrol Opposing Mechanisms of Helicobacter pylori in Gastric Cancer” because this review aims to highlight how resveratrol and H. pylori infection have opposing properties in the development of gastric cancer. While resveratrol possesses anti-inflammatory, antioxidant and antitumor properties, H. pylori infection promotes inflammation, oxidative stress and carcinogenesis. We believe that emphasizing this contrast is crucial for understanding the therapeutic potential of resveratrol against H. pylori-induced gastric cancer.
